# Cardiac Surgery Associated AKI Prevention Strategies and Medical Treatment for CSA-AKI

**DOI:** 10.3390/jcm10225285

**Published:** 2021-11-14

**Authors:** Marlies Ostermann, Gudrun Kunst, Eleanor Baker, Kittisak Weerapolchai, Nuttha Lumlertgul

**Affiliations:** 1Department of Critical Care, King’s College London, Guy’s & St Thomas’ NHS Foundation Trust, London SE1 7EH, UK; Eleanor.Baker@gstt.nhs.uk (E.B.); Nuttha.Lumlertgul@gstt.nhs.uk (N.L.); 2Department of Anaesthesia, School of Cardiovascular Medicine & Sciences, King’s College Hospital, King’s College London, London SE5 9RS, UK; gudrun.kunst@kcl.ac.uk; 3British Heart Foundation Centre of Excellence, King’s College London, London SE5 9RS, UK; 4Division of Urology, Department of Surgery, Somdech Phra Pinklao Hospital, Bangkok 10330, Thailand; Kittisak.Weerapolchai@gstt.nhs.uk; 5Department of Urology, Guy’s & St Thomas’ NHS Foundation Trust, London SE1 9RS, UK; 6Department of Internal Medicine and Excellence Center in Critical Care Nephrology, King Chulalongkorn Memorial Hospital Division of Nephrology, Bangkok 10330, Thailand; 7Research Unit in Critical Care Nephrology, Chulalongkorn University, Bangkok 10330, Thailand

**Keywords:** acute kidney injury (AKI), cardiac surgery, cardiac surgery associated AKI, prevention, management, cardiovascular, renal

## Abstract

Acute kidney injury (AKI) is common after cardiac surgery. To date, there are no specific pharmacological therapies. In this review, we summarise the existing evidence for prevention and management of cardiac surgery-associated AKI and outline areas for future research. Preoperatively, angiotensin-converting enzyme inhibitors and angiotensin receptor blockers should be withheld and nephrotoxins should be avoided to reduce the risk. Intraoperative strategies include goal-directed therapy with individualised blood pressure management and administration of balanced fluids, the use of circuits with biocompatible coatings, application of minimally invasive extracorporeal circulation, and lung protective ventilation. Postoperative management should be in accordance with current KDIGO AKI recommendations.

## 1. Background

Acute kidney injury (AKI) is a common complication after cardiac surgery affecting 5 to 42% of patients [1,2,3]. It can develop in the setting of any type of cardiac surgery, including coronary artery bypass graft (CABG) procedures, valve replacement or repair, and complex congenital heart surgery. In its most severe form, AKI is associated with a longer Intensive Care Unit (ICU) and hospital stay, high health care costs, and an increased risk of mortality [1,2,4]. AKI also has an impact on other organs, including the brain, heart, liver, lungs and immune system, which may further increase morbidity and mortality [5,6]. Patients who survive remain at risk of premature chronic kidney disease (CKD), even if renal function initially recovers [7]. Measures to reduce or mitigate the risk of cardiac surgery-associated AKI (CSA-AKI) can have a significant impact in terms of reducing morbidity and healthcare costs and preventing CKD, end-stage kidney disease (ESKD) and long-term cardiovascular disease. Here, we provide a summary of proposed perioperative strategies to reduce the risk of CSA-AKI [1,8,9,10].

### 1.1. Pathophysiology

The pathophysiology of CSA-AKI is multifactorial and includes pre-, intra- and postoperative factors [1,5,11]. Mechanisms include ischaemia–reperfusion injury, renal hypoperfusion, inflammation, oxidative stress, neurohormonal activation, nephrotoxic exposure, raised intra-abdominal pressure (IAP) and renal congestion [1,11]. Cardiopulmonary bypass (CPB), low cardiac output state and bleeding complications may contribute, too. During CPB, blood is exposed to the extracorporeal circuit, which leads to the activation of pro-inflammatory and oxidative stress pathways, as well as an increased production of free radicals. In addition, haemolysis may develop, causing a rise in free haemoglobin and catalytic iron, both of which have direct nephrotoxic effects. Changes in haemodynamics impact the renal microcirculation and may trigger neurohormonal changes, including stimulation of the sympathetic nervous system and activation of the renin–angiotensin–aldosterone system (RAAS). Other potential contributing risk factors are exposure to nephrotoxic agents, septic emboli in the case of infective endocarditis, and cholesterol embolisation.

### 1.2. Aki Diagnosis and Roles of Biomarkers

The diagnosis of CSA-AKI is currently based on traditional markers of renal function, serum creatinine and urinary output. The Kidney Disease: Improving Global Outcomes (KDIGO) criteria define AKI as a rise in serum creatinine by ≥0.3 mg/dl within 48 h, a ≥1.5-fold increase from baseline within 7 days or a reduction in urinary output to <0.5 ml/kg/h for 6 h or more [12,13,14]. However, serum creatinine can be impacted by several non-renal factors, such as muscle mass, hydration and medications that block tubular creatinine secretion [15]. In addition, following a renal insult, a rise in serum creatinine can be delayed by up to 48 h [16]. Similarly, oliguria is not always an indicator of AKI as it can be a physiological response to hypovolaemia. Given these limitations, new biomarkers for the early diagnosis and risk stratification of CSA-AKI have been proposed [17,18,19]. The most widely studied biomarkers are urinary or plasma neutrophil gelatinase-associated lipocalin (NGAL), urinary tissue inhibitor of metalloproteinases-2 (TIMP-2) and insulin-like growth factor binding protein 7 (IGFBP7) [20,21]. Other biomarkers include Dickkopf-3 (DKK3), hepcidin, interleukin-18 (IL-18), kidney injury molecule-1 (KIM-1), liver-type fatty acid-binding protein (L-FABP), N-acetyl-β-D-glucosaminidase (NAG) and proenkephalin-A (penk) [17]. Although this area of research is promising, most biomarkers are not routinely used in clinical practice and more research is necessary to define their roles in predicting, diagnosing and managing CSA-AKI.

## 2. Prevention Strategies

### 2.1. Preoperative Measures

Both pharmacological and non-pharmacological measures have been investigated for the prevention of CSA-AKI [1,9]. Of note, most studies are small, use variable inclusion criteria, apply different doses of medications at different times, investigate variable outcomes and use different criteria to define AKI [2].

#### 2.1.1. Pharmacological Interventions

(i)Corticosteroids

Systemic inflammation during cardiac surgery contributes to AKI and death postoperatively. Corticosteroids can downregulate pro-inflammatory cytokines, upregulate anti-inflammatory cytokines, and have downstream effects that are independent of gene transcription (e.g., stimulation of phosphatidylinositol 3-kinase) [22]. Despite these mechanisms, two large randomised controlled trials (RCTs), “Dexamethasone in Cardiac Surgery” (DECS), which studied 1 mg/kg dexamethaxone as a single dose, and the “Steroids in Cardiac Surgery” (SIRS) trial, which used 500 mg methylprednisolone as a split dose, as well as a subsequent large meta-analysis concluded that steroids had no protective effect against AKI [23,24,25]. However, the main analysis of the DECS trial did not evaluate renal replacement therapy (RRT) as an outcome. In a subsequent post-hoc analysis, the authors reported that dexamethasone was associated with reduced incidence of AKI requiring RRT, especially in those with advanced CKD [26]. Further research is necessary to confirm these findings.

(ii)Albumin

Albumin is the main protein responsible for plasma oncotic pressure. Besides its volume expansion effect, it has protective properties [27]. It binds to toxic endogenous and exogenous compounds, scavenges free radicals, exerts anti-thrombotic effects, acts as a reservoir for nitric oxide and maintains capillary membrane function. A single-centre RCT in patients undergoing off-pump cardiac surgery showed that the correction of hypoalbuminemia (<4 g/dL) by administering 100, 200 or 300 mL of human albumin at 20% according to preoperative serum albumin concentration immediately before surgery was associated with increased urine output during surgery and a reduced risk of postoperative AKI [27]. Further studies are required to confirm these findings.

(iii)Erythropoietin

A porcine model showed that exogenous erythropoetin (EPO) was renoprotective against ischaemia–reperfusion injury via immunomodulatory effects [28]. However, subsequent human trials and a recent meta-analysis concluded that overall, exogenous EPO did not reduce CSA-AKI [29]. However, a subgroup analysis suggested that the dose of EPO may be important: low-dose EPO (200–300 IU/kg) before anaesthesia was protective against CSA-AKI, whereas high-dose EPO (400–500 IU/kg) was not.

(iv)Statins

Statins are 3-hydroxy-3-methyl-glutaryl-CoA reductase inhibitors. Although they are effective at attenuating inflammation and oxidative stress, there is no evidence that their use reduces the incidence of CSA-AKI [30]. In fact, the Statin Therapy in Cardiac Surgery (STICS) trial, which randomised 1922 patients to receive either perioperative rosuvastatin (20 mg daily) or placebo, reported a significantly higher AKI incidence in the treatment group [31]. This finding is yet to be fully explained. The authors speculated that this may be related to the use of rosuvastatin, which has been observed in other contexts to increase the incidence of proteinuria. A study in 44 patients undergoing elective CABG surgery showed no difference in AKI incidence between the simvastatin (20 mg daily) and placebo groups [32]. In contrast, a study in 151 patients undergoing non-coronary artery cardiac surgery showed that serum creatinine was dramatically increased postoperatively in the control group, but not in the simvastatin group (20 mg daily) [33]. Finally, two large meta-analyses including data from observational studies found conflicting evidence regarding the role of preoperative statin in preventing CSA-AKI [34,35], and a Cochrane analysis of small randomized controlled trials found no effect [36]. The AKI guideline of the European Society of Intensive Care Medicine (ESICM) recommends against the perioperative use of high-dose statins in statin-naïve patients to prevent AKI after cardiac surgery [9].

For patients who are on statin therapy before surgery, the results of the STICS trial suggest that discontinuing the medication pre-operatively does not reduce the risk of AKI postoperatively [31]. In this trial, 653 patients were on statin therapy at the time of randomisation; among them, there was no difference in risk of CSA-AKI between patients who continued and those who discontinued the medication before surgery.

(v)N-acetylcysteine

N-acetylcysteine (NAC) is a precursor of intracellular glutathione that possesses antioxidant properties, thereby reducing reactive oxygen species. As oxidative stress plays a role in the pathophysiology of CSA-AKI, the role of NAC has been investigated in several studies. A meta-analysis of 10 studies using NAC at different doses, including 1391 patients, concluded that there was no role for NAC in preventing CSA-AKI [37]. However, the doses used in the different trials varied from 600 mg twice a day to 1200 mg twice a day, 50 mg/kg, 100 mg/kg and 150 mg/kg. The AKI guideline of the ESICM also suggests not using NAC to prevent CSA-AKI [9].

(vi)Sodium bicarbonate

Plasma catalytic iron is the chemical form of iron that catalyses the redox cycle and generates hydroxyl radicals and oxidative damage. It is associated with increased CPB time and the number of red blood cell transfusions, and contributes to the development of CSA-AKI [38]. Free ferric iron precipitates and is excreted as an inert complex in the urine at neutral or alkaline pH. Although sodium bicarbonate increases the renal tubular pH, which limits the generation of hydroxyl radicals and lipid peroxidation [39], studies have failed to show any evidence that sodium bicarbonate is able to reduce the risk of CSA-AKI [40]. In some studies, sodium bicarbonate reduced the need for RRT [41], and in some other trials sodium bicarbonate administration was associated with prolonged mechanical ventilation and longer stay in the ICU [40,42,43]. Of note, the study interventions were similar in all trials, but there were some differences in the total dose of bicarbonate given, ranging from 4 mmol/kg to 5.1 mg/kg over 24 h [41].

(vii)Others

Raised intra-abdominal pressure contributes to the risk of CSA-AKI and should be monitored. If high, strategies should be implemented to relieve the pressure and its impact on the kidneys.

The evidence base regarding continuing versus withholding angiotensin-converting enzyme inhibitors (ACEIs) and angiotensin receptor blockers (ARBs) pre-operatively is limited to observational studies with conflicting data. Most studies reported a higher risk of intraoperative hypotension in patients who had received ACEIs/ARBs pre-operatively; in some studies, an association with postoperative AKI was noted, too [44,45,46]. For this reason, it is currently recommended to withhold ACEIs and ARBs during the perioperative period [1,12]. This includes patients with heart failure. However, evidence-based data to support this recommendation are lacking.

Although cardiac surgery-induced inflammation contributes to the risk of AKI, there is no role for non-steroidal anti-inflammatory drugs (NSAIDs) in preventing CSA-AKI. In fact, NSAIDs may increase the risk of AKI due to reduced prostaglandin synthesis and alterations in renal microcirculation [11]. If aminoglycosides are necessary, they should be used for as little time as possible [47].

(viii)Novel therapies in development

QPI-1002 is a synthetic small interfering ribonucleic acid (si-RNA) designed to temporarily downregulate the expression of the pro-apoptotic gene p53 via the activation of the RNA interference (RNAi) pathway. The inhibition of p53 may provide additional time for renal tubular epithelial cells to repair before the initiation of apoptosis. A phase I study in patients at risk of AKI after on-pump cardiac surgery showed that QPI-1002 was safe and well-tolerated across all doses [48]. The results of phase 3 studies are awaited (NCT03510897, NCT02610283).

Lithium is a potent inhibitor of glycogen synthase kinase 3β (GSK3β), a highly conserved serine/threonine protein kinase that is centrally involved in a number of signalling cascades, including the reperfusion injury salvage kinase pathway (involving PI3 K/Akt signalling) and the survivor activating factor enhancement pathway (involving JAK/STAT3 signalling) [49]. Lithium has been shown to promote renal tubular epithelial survival and the recovery of kidney function in pre-clinical studies [50]. A clinical study randomising patients to receive 3 days of lithium (300 mg) versus placebo in patients undergoing cardiac surgery with CPB is ongoing [51].

#### 2.1.2. Non-Pharmacological Interventions

(i)Intra-Aortic Balloon Pump

There is an association between the use of an intra-aortic balloon pump (IABP) during CPB in selected high-risk patients and improved whole-body perfusion, less endothelial activation, reduced incidence of AKI and reduced need for RRT [52,53]. However, there is some concern that an IABP may also significantly lower aortic pressure in the distal portion of the aorta, and in fact impair renal perfusion [54]. Therefore, it should only be used in carefully selected patients.

(ii)Contrast administration

Contrast exposure increases the risk of CSA-AKI. Cardiac surgery should be delayed until 24 to 72 h after the administration of contrast to reduce the nephrotoxic burden, provided the clinical conditions allow this [1].

### 2.2. Intraoperative Strategies

Several intraoperative measures have been studied to either protect the kidneys or to optimise renal blood flow and oxygen delivery during surgery. These include surgical and anaesthetic techniques, haemodynamic management and fluid therapy.

#### 2.2.1. Surgical Techniques

The question of whether CPB and subsequent systemic inflammation due to blood contact with the foreign CPB circuit have an effect on kidney function has been addressed most extensively in a large multicentre RCT comparing off-pump with on-pump group coronary artery bypass graft (CABG) surgery (*n* = 4752) [55,56]. There was no difference in the incidence of the most severe form of postoperative AKI; the new renal failure requiring dialysis was 1.3% in both groups after 1 year, with a 5-year incidence of 1.7% in the off-pump and 1.9% in the on-pump group. Similar results were seen in two other large multicentre RCTs comparing off-pump and on-pump cardiac surgery [57,58].

#### 2.2.2. CPB-Related Factors

(i)Haemodynamics

Haemodynamic management during CPB is crucial to maintaining adequate perfusion pressure in the kidneys. A recent trial comparing a high mean arterial blood pressure (MAP) target (70–80 mmHg) with a low MAP target (40–50 mmHg) during CPB showed a significantly larger number of patients doubling their baseline creatinine levels in the high target group [59]. To achieve the higher MAP target, more patients received vasopressors in the high-target group. These results suggest that a higher blood pressure during CPB is associated with a higher usage of vasopressors, which increases creatinine levels postoperatively. In contrast, another study in 410 patients undergoing cardiac surgery with CPB demonstrated that blood pressure excursions below the lower limit of the cerebral autoregulation threshold were associated with AKI [60]. This implies that individualised blood pressure management may be most effective. The mean lower limit of cerebral autoregulation (LLCO) is 66 mmHg, but individual LLCOs vary between 40 and 90 mmHg [61]. Currently, the limits of cerebral autoregulation are not routinely assessed during cardiac surgery. Therefore, the recent European guidelines recommend MAP targets between 50 and 80 mmHg during CPB [8].

In addition to blood pressure, pump flow and flow-related oxygen delivery are also important factors. Oxygen delivery during CPB has been shown to be directly associated with postoperative AKI [62]. A large multicentre RCT comparing a goal-directed perfusion strategy, aimed at maintaining a high-normal oxygen delivery target (DO_2_) of above 280 mL·min^−1^·m^−2^, with a conventional perfusion strategy demonstrated that patients receiving goal-directed perfusion during CPB had a significantly lower incidence of postoperative AKI [62]. Although it is common practice to accept non-pulsatile flow during CPB, it has been suggested that more physiological pulsatile perfusion techniques may be more protective for kidney function and result in better postoperative creatinine clearance [63].

An increase in IAP can lead to decreased renal blood flow and glomerular filtration rate [64]. A number of factors can increase IAP in patients undergoing cardiac surgery; these include obesity, duration of CPB, vasopressor use and use of red blood cell transfusion [65]. It has been reported that 27–83% of patients undergoing cardiac surgery have raised IAP. Recognising risk factors for raised IAP and ensuring the monitoring of IAP in high-risk patients may prevent deleterious effects, including AKI.

(ii)Biocompatible coatings

Heparin-bonded or phosphorylcholine-coated circuits have been proposed to provide a more physiological surface, thereby reducing the risk of haemolysis and the activation of pro-inflammatory and oxidative stress pathways. Studies investigating their role have confirmed improved outcomes, such as reduced requirements for blood transfusions and shorter lengths of stay in hospital, but also reduced elevation of creatinine postoperatively and potential renal protection [66,67,68]. Furthermore, recent European guidelines have recommended biocompatible coatings with class IIa evidence, i.e., despite conflicting evidence, the weight of evidence is considered to favour their usefulness [8].

(iii)Minimally invasive extracorporeal circulation

The concept of minimally invasive extracorporeal circulation (MiECC) includes the routine use of heparin-coated circuits, a small priming volume, a closed system, a centrifugal pump and no venous reservoir. A retrospective propensity score-matched analysis showed that MiECC was associated with a lower incidence of AKI post CABG surgery [69]. However, a subsequent small RCT comparing miniaturized with conventional CPB in 68 patients undergoing cardiac surgery demonstrated no difference in the incidence of AKI [70]. This may be due to the fact that some of the minimally-invasive techniques, such as minimal priming volumes and heparin-coated circuits, have already been adopted into routine conventional CPB in the last decade.

(iv)Leucocyte depletion

In order to attenuate inflammation, leucocyte depletion (LD) filters have been studied. Experimental data showing that neutrophils and leucocytes accumulate in the kidneys following ischaemia–reperfusion-mediated injury provide a potentially plausible reason for the usage of LD filters to prevent AKI [71]. Interestingly, a meta-analysis including six trials and 374 patients found that LD filters reduced the incidence of worsening kidney function [72]. However, the sample size of individual trials was very small, and there were varying definitions of AKI.

A recent Cochrane review found no benefit when other clinically relevant outcomes, such as length of ICU and hospital stay, were assessed [73]. Furthermore, the most recent European CPB guidelines do not recommend the routine use of LD [8].

(v)Duration of CPB

The duration of CPB and aortic cross clamp times and the related longer exposure to CPB-related inflammation are well known to be directly correlated with an increased risk of AKI [74,75].

#### 2.2.3. Anaesthesia-Related Techniques

(i)Remote ischaemic preconditioning

Remote ischaemic preconditioning (RIPC) is a technique that involves repetitive brief periods of ischemia induced by inflating a blood pressure cuff in the upper arm or thigh for several minutes, and then releasing the cuff to allow reperfusion of distal tissues. Although the mechanism is not well understood, it is proposed that the ischaemia–reperfusion process stimulates the release and activation of anti-inflammatory cytokines, hypoxia-inducible factor (HIF) and neural autonomic and humoral signalling pathways, thus preventing and attenuating distal organ dysfunction [76]. The role of RIPC in preventing CSA-AKI is controversial. Whereas experimental data and results from small RCTs support the application of RIPC, large multicentre RCTs have found no difference in the incidence of postoperative AKI [77,78,79]. However, it is possible that RIPC might be beneficial in the prevention of AKI in specific subgroups of patients [80,81].

(ii)Volatile versus intravenous anaesthesia

Large amounts of experimental data suggest that volatile anaesthetics (i.e., desflurane, isoflurane and sevoflurane) have renoprotective effects [82]. However, the intravenous anaesthetic propofol may also protect the kidneys, as demonstrated in a small RCT in patients undergoing valve surgery [83]. The largest clinical study comparing volatile with intravenous anaesthesia, including 5400 patients, demonstrated similar postoperative mortality rates with either technique [84]. Furthermore, a subsequent meta-analysis comparing volatile and intravenous anaesthesia did not show any difference in postoperative AKI [85]. In summary, the potential renal protective effect of either volatile anaesthetics or intravenous anaesthetics, as demonstrated in experimental research, has not been consistently shown in large clinical trials.

Dexmedetomidine is an alpha-2-adrenoreceptor agonist that exerts sedative, analgesic and sympathicolytic effects [86]. The administration of dexmedetomidine immediately after induction or before CPB might attenuate inflammatory cytokines, danger-associated molecular patterns and renal ischaemia [87]. A recent meta-analysis of nine RCTs including 1308 patients and using a range of dexmedetomidine loading doses (0.4–1 µg/kg) and rates of continuous infusion (0.04–0.6 µg/kg/h) reported a significantly reduced incidence of CSA-AKI, particularly in patients older than 60 years [88,89]. Larger high-quality trials are required to confirm this finding and to identify the optimal dose.

#### 2.2.4. Intraoperative Fluid and Goal Directed Therapy

(i)Fluids

A single-centre prospective cohort study indicated that a higher positive fluid balance was associated with an increased requirement for RRT following cardiac surgery [90]. Similarly, a large retrospective analysis of cardiac surgery demonstrated an association between a positive fluid balance and AKI [91]. However, the role of fluid restriction in cardiac surgery has not yet been investigated [92,93].

The type of fluid also plays a role in relation to the risk of postoperative AKI. A meta-analysis in the general perioperative setting, including 21 studies and 6253 patients undergoing surgery or admitted to the ICU, concluded that the administration of fluids containing lower chloride concentrations was associated with a significantly lower risk of AKI [94]. This finding was confirmed in a large RCT including 600 patients undergoing off-pump CABG surgery, where the use of balanced crystalloid solutions significantly decreased stage 1 AKI postoperatively [95].

(ii)Goal-directed haemodynamic therapy

Goal-directed therapy (GDT) is a strategy to increase cardiac output by using fluids and/or inotropes to improve oxygen delivery to organs and peripheral tissues. GDT has been shown to be renoprotective, resulting in a lower incidence of AKI after cardiac surgery [96,97]. Furthermore, GDT during cardiac surgery has been recommended as a class I recommendation (i.e., with evidence and/or general agreement that this procedure is beneficial, useful and effective) in recent European guidelines [8].

(iii)Blood products

Preoperative anaemia has been associated with AKI and mortality after cardiac surgery, but the transfusion of two or more units of packed red blood cells during surgery is also considered a risk factor for CSA-AKI [98,99]. Two large RCTs, in which patients were randomized to a liberal versus restrictive red cell transfusion strategy, intraoperatively and postoperatively (haemoglobin trigger <9.5 g/dL versus <7.5 g/dL), showed no difference in postoperative AKI [100,101]. Thus, blood transfusion beyond traditional transfusion triggers is not considered an effective strategy to protect kidney function.

#### 2.2.5. Mechanical Ventilation

The receipt of mechanical ventilation is a common risk factor for AKI during critical illness, particularly in patients with acute respiratory failure. The contributing mechanisms include haemodynamic, neurohormonal and immune-mediated processes. Mechanical ventilation strategies during CPB were investigated in a meta-analysis and semi-quantitative review of 16 RCTs, including 814 patients [102]. Whilst continuous positive airway pressure (CPAP) and vital capacity manoeuvres during CPB improved oxygenation variables directly after CPB, there was no sustained benefit, and AKI was not included as an outcome variable. Whether particular ventilation strategies during cardiac surgery (pre- and post-CPB) protect kidney function more than others remains unclear. Until further research results are available, it is recommended to monitor tidal volumes and ventilation pressures and to apply lung-protective ventilation strategies whenever possible, in order to reduce the impact on lung but also kidney function [103].

#### 2.2.6. Drugs

There are no specific pharmacological interventions to prevent CSA-AKI. Many drugs, including dopamine, diuretics, mannitol, and natriuretic peptides, have been studied. Although they may increase urine output, none are routinely used due to limited and conflicting data and, in some cases, evidence of harm [1].

Mannitol: Mannitol is an osmotic diuretic that is often used in the priming fluid for CPB. The use of mannitol was investigated in two small RCTs in 50 patients with established renal dysfunction and 40 patients with normal preoperative kidney function [104,105]. Both trials used 0.5 g/kg of mannitol in the pump prime. No renoprotective effect of mannitol was demonstrated.

Furosemide: Furosemide is a loop diuretic that is widely used to prevent or manage fluid overload. The prophylactic administration of furosemide in different doses, given intermittently as boluses or continuous infusion, was not shown to reduce postoperative AKI after cardiac surgery in two small RCTs, and therefore is also not recommended [106,107].

Atrial natriuretic peptide (ANP): ANP has been reported to inhibit the RAAS and sympathetic nervous system, both of which contribute to the pathophysiology of CSA-AKI. Meta-analyses of studies performed in patients undergoing cardiovascular surgery showed that there was a reduced need for RRT with low-dose ANP (<100 ng/kg/min) [108]. Conversely, in high doses (>100 ng/kg/min), ANP was associated with more adverse events. However, the majority of studies investigating ANP were underpowered, and considered of low or moderate quality. Therefore, ANP is not currently recommended for the treatment of AKI [1].

Fenoldopam: Fenoldopam is a selective agonist of dopamine D1 receptors. It causes the relaxation of smooth muscles, vasodilation, and the inhibition of sodium reabsorption in the renal tubules. Fenoldopam has also been studied in several trials [109,110]. Although meta-analyses have suggested a decrease in RRT in patients with CSA-AKI who were treated with fenoldopam at different doses (0.1–0.3 mg/kg/min for a range of 12–96 h), a multicentre RCT in patients with CSA-AKI (*n* = 667) was stopped for futility after an interim analysis [110,111]. Compared with placebo, fenoldopam infusion at 0.025–0.3 µg/kg/min for up to 4 days did not reduce the need for RRT, but caused more harm, in particular causing hypotension.

### 2.3. Postoperative Strategies

The prospect of identifying patients with early kidney injury after surgery prior to a rise in serum creatinine or fall in urine output offers opportunities for interventions to mitigate further damage [17]. The 2012 KDIGO guideline recommends a variety of supportive measures, including the optimisation of fluid status and haemodynamics, the early consideration of functional haemodynamic monitoring, the avoidance of hyperglycaemia and radiocontrast agents, and the discontinuation of nephrotoxic medications if possible [12] (Table 1).

A recent observational study including 12 hospitals in different European countries showed that in routine clinical practice, all components of the KDIGO recommendations were only followed in 5.3% of patients after cardiac surgery [112]. On average, each high-risk patient received 3.4 of the 6 measures. Two RCTs investigated whether the implementation of the KDIGO recommendations impacted the occurrence of CSA-AKI in high-risk patients [20,21]. In both studies, the urinary cell cycle arrest biomarkers TIMP-2 and IGFBP7 were measured 4 h after cardiac surgery to identify patients with early kidney injury [113]. The first RCT was a single-centre study of 276 patients that showed that adherence to the KDIGO recommendations for 12 h after a positive urinary cell cycle arrest marker result led to a significantly lower incidence and reduced the severity of AKI compared to patients in the control group [21]. However, this did not translate to a significant reduction in any of the other secondary outcomes, including all-cause mortality, requirement for RRT, and length of stay in ICU or hospital. The second RCT was a multi-national study of similar design, including the same definition of high risk and implementation of the same KDIGO care bundle [20]. It showed that compliance with the KDIGO bundle was 61.2% higher in the intervention group than the control group, and there was a significant reduction in stage 2–3 AKI in the intervention group. However, there was no significant difference in overall rates of AKI or any of the secondary outcomes, including mortality, renal recovery, length of stay in ICU or hospital and persistent renal dysfunction at day 90. Outcomes beyond 90 days, including the risk of premature CKD, were not explored. The guidelines for perioperative care in cardiac surgery by the Enhanced Recovery After Surgery Society recommend the routine use of urinary cell cycle arrest biomarkers after cardiac surgery to identify patients for intensified management according to the KDIGO AKI guideline [10].

Fluid therapy after cardiac surgery is part of haemodynamic management. The administration of sodium bicarbonate after cardiac surgery is not effective at preventing AKI, based on a meta-analysis of three RCTs including 877 patients [41]. However, sodium bicarbonate treatment was associated with a reduced need for RRT in elective coronary artery bypass patients.

## 3. Limitations and Research Recommendations

Although there has been promising research in the areas of preventing and managing CSA-AKI, there are limitations to the aforementioned studies. Firstly, there is a wide disparity in the definitions used for AKI, which makes it difficult to compare data between studies. Baseline CKD reflects reduced renal function reserve, and is an important predictor of post-operative renal outcomes. Therefore, any intervention may have a larger impact in this subgroup of patients. However, most studies include participants with a range of baseline renal function, and therefore, this impact may be masked. Research has focussed on CSA-AKI following CABG and valvular surgery, with few data on the effects of other types of cardiac surgery. Additionally, it is emerging that different phenotypes of AKI exist with different underlying pathophysiological mechanisms and prognoses [114]. To date, this has not been addressed in research studies in the setting of cardiac surgery. Finally, only a small number of studies include AKI as a primary outcome, meaning this is not the focus of the majority of research.

Further research is required to specifically assess CSA-AKI in high-risk subgroups (those with reduced baseline renal function) following a wider variety of cardiac surgeries. In addition, larger clinical trials are required to determine the utility and cost-effectiveness of novel biomarkers for early diagnosis, prognostication and treatment guidance, as well as the efficacy of a number of proposed interventions, such as albumin, erythropoietin, corticosteroids and dexmedetomidine, and novel treatments such as lithium and HIF-stabilizers (e.g., roxadustat, vadadustat, daprodustat, molidustat). A further focus should be other routinely used cardiovascular medications (e.g., aspirin, clopidogrel, nitrates). Lastly, it is important to explore how to implement proven strategies in routine clinical practice and improve adherence to guidelines [112].

## 4. Conclusions

Cardiac surgery-associated AKI is common and associated with high morbidity and mortality. Strategies to prevent or mitigate this complication pre-, intra- or postoperatively are limited to general supportive measures (Figure 1). However, only a few studies have used AKI as the primary outcome, and many heterogeneous definitions of AKI were used, often lacking consensus with KDIGO criteria for AKI [2]. Current evidence supports a multimodal risk-stratification approach, including goal-directed perfusion, the use of biocompatible coatings during CPB, perioperative GDT, the use of balanced fluids with restricted chloride content, and biomarker-guided postoperative management of high-risk patients based on the recommendations of the KDIGO group. If AKI occurs, the transfer of information to all caregivers, medication reconciliation and patient education are essential to reduce the risk of long-term complications [115].

## Figures and Tables

**Figure 1 jcm-10-05285-f001:**
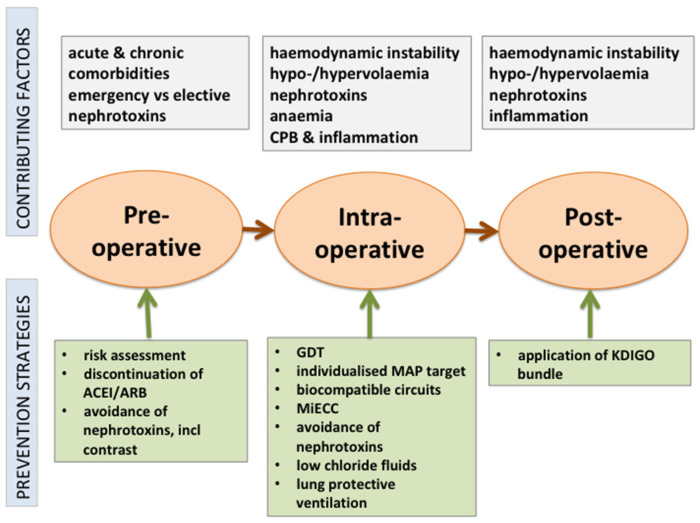
Perioperative strategies to prevent AKI after cardiac surgery.

**Table 1 jcm-10-05285-t001:** KDIGO bundle recommendation and their rationale.

Component of KDIGO Care Bundle	Rationale for Nephroprotection
Close monitoring of renal function	Close and regular monitoring is required to ensure early the diagnosis of AKI. Whilst the exact intervals for checking serum creatinine are not defined, close monitoring of renal function after cardiac surgery is strongly recommended.
Functional haemodynamic monitoring	Several studies have confirmed that cardiac output, systemic oxygen delivery, central venous pressure and systemic haemodynamics are associated with risk of AKI. Functional haemodynamic monitoring allows the early identification of high-risk patients and opportunities for early intervention.
Optimising fluid status and haemodynamics	Both severity and duration of intraoperative hypotension are strong risk factors for postoperative AKI. Similarly, both hypo- and hypervolaemia are associated with an increased risk of AKI. Although the target blood pressure in individual patients and the optimal type and volume of fluid to prevent AKI have not been identified yet, the detection of hypotension and hypo- or hypervolaemia should prompt immediate resuscitation.
Avoidance of hyperglycaemia	Uncontrolled hyperglycaemia significantly increases the incidence of AKI post cardiac surgery via multiple pathways, including the increased production of inflammatory cytokines, the overproduction of superoxide by the mitochondrial electron transport chain, and osmotic diuresis. Conversely, intensive glycaemic control can have adverse effects.
Avoidance of radiocontrast and discontinuation of nephrotoxic medications	Many medications, including radiocontrast media, are risk factors for the development of AKI due to mechanisms such as reduced glomerular perfusion, tubular cell toxicity, intratubular crystal formation and the alteration of the tubular microcirculation.

Abbreviations: AKI = acute kidney injury; KDIGO = Kidney Disease: Improving Global Outcomes. Information adapted from Kellum, J.A. et al [12].

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
