# Peer review of "Cardiac Surgery Associated AKI Prevention Strategies and Medical Treatment for CSA-AKI"

_jcm, 2021, doi:10.3390/jcm10225285_

Round 1
Reviewer 1 Report
In this review article, Dr. Ostermann and colleagues discussed the determinants of AKI in cardiac surgery from pre-, intra and post-operative aspects and reviewed the current evidence and possible interventions to limit AKI in cardiac surgery. Overall, the manuscript is nicely written and enjoyable to read. The content could potentially be useful in the clinical setting. However, several information is missing and needs to be added. Moreover, some parts need improvement to increase clarity:
- The abstract is a bit weird. I feel that this sentence is meaningless "In this review, we summarise the existing evidence and outline areas for future research." because the authors have outlined the main idea of the review prior to this sentence. Maybe this sentence can be put a bit earlier (before summarizing the content).
- The background needs a lot of improvement. It is too short and brief. The authors are suggested to add these components:
- Please add the epidemiology (e.g., prevalence) of AKI in cardiac surgery. How common is "common" as mentioned in line 28?
- When mentioning cardiac surgery in this manuscript, is this limited to CABG or other types of cardiac surgery, such as valve replacement, congenital defect repair etc.? Please explain this important aspect more clearly.
- "The pathophysiology is multifactorial", please extensively elaborate the pathophysiology of AKI in cardiac surgery.
- Line 30: "non-renal organ dysfunction", which organs? Please specify.
- "Patients who survive remain at risk of premature chronic kidney disease (CKD), even if renal function initially recovers." Please add the citation to this statement.
- Line 34: "long-lasting impact" what does it mean?
- Line 37: the section about "preoperative measures" was not at the same level as "B" and "C". Please check again.
- The subheading about ACEi/ARB has to be clearer. At first, it seems to me that those drugs are recommended to limit AKI. Maybe the title should be "Limitation of ACEi/ARB" or something else.
- What if the patients have heart failure? Should ACEi/ARB be terminated before surgery? Please discuss.
- Lines 77-79: Please speculate on the cause why statins increased AKI in STICS trial? What if the patients already consume statins due to prior CVDs or dyslipidemia? should it be terminated before surgery? How long before?
- Regarding N-acetylcysteine, please put more context on why it is relevant to be discussed? Is it common to be used pre-surgery? What for?
- Why NSAIDs should be avoided, while corticosteroid could be beneficial? Please explain.
- In general, please discuss the (potential) underlying mechanisms of each verdict. This approach would be more insightful rather than just reporting the result of previous studies.
- Also, please discuss drugs that are commonly used before cardiac surgery, for example clopidogrel, opioid, nitrates, even oxygen and other cardiovascular drugs.
- Line 124: "Cardiac surgery should be delayed until 24 to 72 hours after the administration of contrast provided the clinical condition allows", Why? please elaborate. How the delay of surgery could help reducing the AKI-inducing effect of contrast?
- Line 131: "subsequent systemic inflammation due to blood contact with the foreign CPB circuit", through which mechanisms? Please elaborate.
- Lines 196-198: "It is not surprising that the duration of CPB and aortic cross clamp times and the related longer exposure to CPB-related inflammation are directly correlated with risk of AKI" Why is it not surprising? Please explain.
- In this particular context "Volatile versus intravenous anaesthesia", please always mention the name of the drugs used in the studies / discussed because each drug can have different properties and they cannot be generalized as volatile vs. iv, unless the study compared many drugs within the same group / category.
- Is there any potential prophylactic approach to reduce AKI in cardiac surgery other than remote conditioning? Maybe like pharmacological agents that can prevent AKI?
- What about laboratory markers? Is there any predictive marker of AKI in cardiac surgery?
- In the abstract, the authors said that they also outlined areas for future research but this was not easy to extract from the manuscript. I would suggest the authors to dedicate a separate section discussing this component (maybe before conclusion).
Author Response
- In this review article, Dr. Ostermann and colleagues discussed the determinants of AKI in cardiac surgery from pre-, intra and post-operative aspects and reviewed the current evidence and possible interventions to limit AKI in cardiac surgery. Overall, the manuscript is nicely written and enjoyable to read.
Response: Thank you for your comments.
- The content could potentially be useful in the clinical setting. However, several information is missing and needs to be added. Moreover, some parts need improvement to increase clarity.
The abstract is a bit weird. I feel that this sentence is meaningless "In this review, we summarise the existing evidence and outline areas for future research." because the authors have outlined the main idea of the review prior to this sentence. Maybe this sentence can be put a bit earlier (before summarizing the content).
Response: We thank the reviewer for their suggestions. We have moved the sentence as suggested. We hope that the revised paper provides more clarity.
- The background needs a lot of improvement. It is too short and brief. The authors are suggested to add these components:
Please add the epidemiology (e.g., prevalence) of AKI in cardiac surgery. How common is "common" as mentioned in line 28?
Response: As suggested, we have expanded the background. We have included prevalence data and added a separate paragraph describing the pathophysiology of cardiac surgery associated AKI. (page 3)
- When mentioning cardiac surgery in this manuscript, is this limited to CABG or other types of cardiac surgery, such as valve replacement, congenital defect repair etc.? Please explain this important aspect more clearly.
Response: We have explained that cardiac surgery includes different types of surgery (page 3).
- "The pathophysiology is multifactorial", please extensively elaborate the pathophysiology of AKI in cardiac surgery.
Response: We have added a paragraph summarising the pathophysiology. (page 3)
- Line 30: "non-renal organ dysfunction", which organs? Please specify.
Response: The sentence has been revised as requested. (page 3)
- "Patients who survive remain at risk of premature chronic kidney disease (CKD), even if renal function initially recovers." Please add the citation to this statement.
Response: A reference has been added.
- Line 34: "long-lasting impact" what does it mean?
Response: We have replaced 'long-lasting' with 'significant' and expanded on the impact.
- Line 37: the section about "preoperative measures" was not at the same level as "B" and "C". Please check again.
Response: Thank you for this comment. We confirm that all headings are now at the same level.
- The subheading about ACEi/ARB has to be clearer. At first, it seems to me that those drugs are recommended to limit AKI. Maybe the title should be "Limitation of ACEi/ARB" or something else.
Response: We apologise for being unclear. We agree that it may be misleading to start with medications that should be discontinued. We have therefore moved the section related to ACEi/ARBs to the paragraph "Others". (page 6) We have also made it clearer that they should be withheld.
- What if the patients have heart failure? Should ACEi/ARB be terminated before surgery? Please discuss. Response: There are no studies looking at a subgroup of patients with heart failure specifically. Therefore, there is insufficient evidence to support continuing or discontinuing ACEi/ARBs in this patient cohort. We have added a sentence to this effect. (page 5)
- Lines 77-79: Please speculate on the cause why statins increased AKI in STICS trial? What if the patients already consume statins due to prior CVDs or dyslipidemia? should it be terminated before surgery? How long before?
Response: We thank the authors for these questions. The authors of the STICS trial speculated that the higher incidence of AKI observed in the treatment group may relate to the use of rosuvastatin which has been observed in other contexts to increase the incidence of proteinuria. For patients already on statin therapy before surgery, the STICS trial suggests that discontinuing the medication pre-operatively does not reduce the risk of AKI postoperatively. In this trial, 653 patients were on statin therapy at the time of randomization; among them, there was no difference in risk of CSA-AKI between patients who continued and those who discontinued the medication before surgery. Therefore, discontinuing statins before cardiac surgery does not reduce the risk of AKI.
We have revised the paragraph and incorporated this information. We also added more background information and mentioned a few more studies, two meta-analyses and a Cochrane review. (page 5)
- Regarding N-acetylcysteine, please put more context on why it is relevant to be discussed? Is it common to be used pre-surgery? What for?
Response: Thank you for this comment. N-acetylcysteine (NAC) is a precursor of intracellular glutathione which possesses antioxidant properties. As oxidative stress plays a role in the pathophysiology of CSA-AKI, the role of NAC has been investigated in several studies. A meta-analysis of 10 studies including 1391 patients concluded that there was no role for NAC in preventing CSA-AKI. Hopefully it is no longer used pre-surgery but it was considered a potential drug for prevention of AKI. Hence we felt that it should be included in this review.
- Why NSAIDs should be avoided, while corticosteroid could be beneficial? Response: We thank the reviewer for this question. Cardiac surgery induced inflammation contributes to the risk of AKI. As a result, various drugs with anti-inflammatory properties have been studied to prevent AKI. Whilst corticosteroids could be beneficial, NSAIDs are potentially harmful due to their inhibitory effect on prostaglandin synthesis and the impact on renal microcirculation.
- In general, please discuss the (potential) underlying mechanisms of each verdict. This approach would be more insightful rather than just reporting the result of previous studies.
Response: As suggested, we have added information about the rationale and potential underlying mechanisms of various drugs and strategies.
- Also, please discuss drugs that are commonly used before cardiac surgery, for example clopidogrel, opioid, nitrates, even oxygen and other cardiovascular drugs.
Response: The reviewer raises an important point. Unfortunately, there are no data regarding other cardiovascular drugs and their role in preventing CSA-AKI. We have referred to this lack of research in the research recommendations section. (page 14)
- Line 124: "Cardiac surgery should be delayed until 24 to 72 hours after the administration of contrast provided the clinical condition allows", Why? please elaborate. How the delay of surgery could help reducing the AKI-inducing effect of contrast?
Response: In general, the number of potentially nephrotoxic exposures correlates with the risk of CSA-AKI. Patients undergoing cardiac surgery are at risk of AKI due to CPB-related factors, including inflammation, haemolysis and haemodynamic instability. Additional contrast exposure increases the risk further. If the patient's condition allows, surgery should be delayed until the effects of contrast have subsided. However, we acknowledge that cardiac surgery might be life-saving and can't always be delayed. We have rephrased the statement. (page 7)
- Line 131: "subsequent systemic inflammation due to blood contact with the foreign CPB circuit", through which mechanisms? Please elaborate.
Response: We have added a paragraph regarding the pathophysiology of CSA-AKI and described that blood contact with the circuit leads to the activation of pro-inflammatory pathways and complement. In addition, CPB can induce haemolysis and trigger the production of free radicals and catalytic iron which contribute to the development of AKI. (page 3)
- Lines 196-198: "It is not surprising that the duration of CPB and aortic cross clamp times and the related longer exposure to CPB-related inflammation are directly correlated with risk of AKI" Why is it not surprising? Please explain.
Response: Several previous studies have shown that the duration of CPB and aortic cross clamp time are associated with risk for CSA-AKI. We have added an additional reference. (page 10)
- In this particular context "Volatile versus intravenous anaesthesia", please always mention the name of the drugs used in the studies / discussed because each drug can have different properties and they cannot be generalized as volatile vs. iv, unless the study compared many drugs within the same group / category.
Response: We have included examples of volatile anesthesia in the paragraph (desflurane, isoflurane and sevoflurane). The mentioned RCT grouped all drugs in the same category (sevoflurane (83.2%], desflurane [9.2%] and isoflurane [5.8%]. In the referenced meta-analysis, volatile anaesthetics were analysed as a class for their effects on AKI/renal replacement therapy.
- Is there any potential prophylactic approach to reduce AKI in cardiac surgery other than remote conditioning? Maybe like pharmacological agents that can prevent AKI?
Response: We thank the reviewer for this important question. To date, several agents (for example, dexmedetomidine, erythropoietin) have shown promising results in small trials for the prevention of CSA-AKI. Larger trials need to be conducted to confirm these results. We have added this in the research recommendations section. (page 14)
- What about laboratory markers? Is there any predictive marker of AKI in cardiac surgery?
Response: We thank the reviewer for the suggestion. We have added a section regarding AKI diagnosis and biomarkers. (pages 3 and 4)
- In the abstract, the authors said that they also outlined areas for future research but this was not easy to extract from the manuscript. I would suggest the authors to dedicate a separate section discussing this component (maybe before conclusion).
Response: We thank the reviewer for this suggestion and have added a paragraph outlining recommendations for future research. (page 14)
Reviewer 2 Report
This is an excellent and thorough review. It is principally focuses on clinical data. Yet, I propose to include a brief review on the pathophysiology, at least regarding renal hypoxic injury, differing the impact of total cessation of RBF from that caused by ongoing renal hypothermic perfusion at lower pressures. This is highly relevant to the cardiac surgery scenario with diverse patterns of tubular damage (Kidney Int. 2010 Jan;77(1):9-16). Such a short summary might also provide insights regarding potential interventions and precautions (such as avoidance of NSAIDs or radiocontrast studies, or the potential impact of furoseminde, mannitol or ANP).
At least on physiologic grounds, I would recommend avoiding increased intra-abdominal pressure following surgery, aimed at reducing the risk of intensified renal parencymal hypoxia (J Card Fail. 2019 Jun;25(6):468-478; J Int Med Res 2021 May;49(5): 3000605211016627).
The authors should generally address an important methodological issue in the assessment of the value of protective interventions, namely patients' selection in RCS. Baseline renal failure is the most important predictor of post-op renal outcome, aside from perioperative parameters. As renal failure at baseline reflects reduced renal functional reserve (Clin Exp Pharmacol Physiol. 2021 Sep 20. doi: 10.1111/1440-1681), studies focusing on this particular group of patients might better show an impact of an intervention. Unfortunately, most if not all studies include heterogeneous groups of patients regarding baseline GFR, hence reduce the likelihood to detect an effect. Let us take the NAC story, for instance. The authors briefly state that based on a recent meta-analysis there is no evidence for the beneficial impact of N acetylcystein in the prevention of post op AKI. Nevertheless, they ignore the observation revealed in this meta-analysis, indicating that the likelihood to show renal protection with NAC was significantly higher at lower baseline kidney function. This trend has already been suggested in the first RCS published on NAC in cardiac surgery (Burns, JAMA 2005). The diluting effect of patients with intact kidneys at baseline should be taken into account in all other sections of the manuscript, addressing how many patients in the various reports were indeed at risk with impaired renal function at baseline
The authors should also address the potential importance of using additional indicators of renal injury for the determination of an impact of intervention, namely biomarkers of renal tubular injury. This might be especially important in the detection of sub-clinical AKI (J Clin Med. 2021 May 14;10(10):2120).
Following the section dealing with RIPC, you might address the possible role of HIF in RIPC and the potential use of HIF-prolyl hydroxylase inhibitors (Acta Physiol (Oxf). 2016 Apr;216(4):395-406) for both renal and cardiac protection. This might be followed by the use of EPO, indeed a HIF-target gene

Author Response
- This is an excellent and thorough review. It is principally focuses on clinical data. Yet, I propose to include a brief review on the pathophysiology, at least regarding renal hypoxic injury, differing the impact of total cessation of RBF from that caused by ongoing renal hypothermic perfusion at lower pressures. This is highly relevant to the cardiac surgery scenario with diverse patterns of tubular damage (Kidney Int. 2010 Jan;77(1):9-16). Such a short summary might also provide insights regarding potential interventions and precautions (such as avoidance of NSAIDs or radiocontrast studies, or the potential impact of furoseminde, mannitol or ANP).
Response: We thank the reviewer for the suggestion. We have added a paragraph outlining the pathophysiology of CSA-AKI. (page 3)
- At least on physiologic grounds, I would recommend avoiding increased intra-abdominal pressure following surgery, aimed at reducing the risk of intensified renal parencymal hypoxia (J Card Fail. 2019 Jun;25(6):468-478; J Int Med Res 2021 May;49(5): 3000605211016627).
Response: We thank the reviewer for this suggestion and agree. We have mentioned 'raised intra-abdominal pressure' as a risk factor for AKI after cardiac surgery (page 3) and included a recommendations to monitor intra-abdominal pressure in high-risk patients. (page 6)
- The authors should generally address an important methodological issue in the assessment of the value of protective interventions, namely patients' selection in RCS. Baseline renal failure is the most important predictor of post-op renal outcome, aside from perioperative parameters. As renal failure at baseline reflects reduced renal functional reserve (Clin Exp Pharmacol Physiol. 2021 Sep 20. doi: 10.1111/1440-1681), studies focusing on this particular group of patients might better show an impact of an intervention. Unfortunately, most if not all studies include heterogeneous groups of patients regarding baseline GFR, hence reduce the likelihood to detect an effect. Let us take the NAC story, for instance. The authors briefly state that based on a recent meta-analysis there is no evidence for the beneficial impact of N acetylcystein in the prevention of post op AKI. Nevertheless, they ignore the observation revealed in this meta-analysis, indicating that the likelihood to show renal protection with NAC was significantly higher at lower baseline kidney function. This trend has already been suggested in the first RCS published on NAC in cardiac surgery (Burns, JAMA 2005). The diluting effect of patients with intact kidneys at baseline should be taken into account in all other sections of the manuscript, addressing how many patients in the various reports were indeed at risk with impaired renal function at baseline
Response: We thank the reviewer for raising this important point. As suggested, we have included a paragraph outlining some limitations of current studies. (page 14)
- The authors should also address the potential importance of using additional indicators of renal injury for the determination of an impact of intervention, namely biomarkers of renal tubular injury. This might be especially important in the detection of sub-clinical AKI (J Clin Med. 2021 May 14;10(10):2120).
Response: We thank the reviewer for this suggestion and have added a paragraph dedicated to biomarkers. (pages 3 and 4)
- Following the section dealing with RIPC, you might address the possible role of HIF in RIPC and the potential use of HIF-prolyl hydroxylase inhibitors (Acta Physiol (Oxf). 2016 Apr;216(4):395-406) for both renal and cardiac protection. This might be followed by the use of EPO, indeed a HIF-target gene.
Response: We thank the reviewer for this suggestion. We have added the suggested section and reference in the manuscript.
Round 2
Reviewer 1 Report
Thank you for addressing my previous comments and suggestions. I do still have two minor suggestions:
- Please use the MDPI template with line number (as done in the first version) to make it easier in providing comments.
- Whenever providing study-based recommendations or objections regarding drugs, please make sure to add the doses because dose-dependence effects are common and crucial. Even in case of the absence of dose-dependence, the authors could say that the effect was seen in all studied doses and mention the range.
Author Response
Thank you for addressing my previous comments and suggestions. I do still have two minor suggestions:
- Please use the MDPI template with line number (as done in the first version) to make it easier in providing comments.
Response: We apologise for this oversight and have added the line numbers.
- Whenever providing study-based recommendations or objections regarding drugs, please make sure to add the doses because dose-dependence effects are common and crucial. Even in case of the absence of dose-dependence, the authors could say that the effect was seen in all studied doses and mention the range.
Response: Thank you for this suggestion. The relevant doses have been added.
Additional comment
We also noted a few spelling and grammar mistakes which have been corrected.
All new changes are marked in red.
Reviewer 2 Report
The authors appropriately addressed my comments
Author Response
Thank you.